# MobTCast: Leveraging Auxiliary Trajectory Forecasting for Human Mobility Prediction

**Hao Xue**
School of Computing Technologies
RMIT University
Melbourne, Australia
`hao.xue@rmit.edu.au`

**Flora D. Salim**
School of Computing Technologies
RMIT University
Melbourne, Australia
`flora.salim@rmit.edu.au`

**Yongli Ren**
School of Computing Technologies
RMIT University
Melbourne, Australia
`yongli.ren@rmit.edu.au`

**Nuria Oliver**
ELLIS Unit Alicante Foundation
Alicante, Spain
`nuria@alum.mit.edu`

## Abstract

Human mobility prediction is a core functionality in many location-based services and applications. However, due to the sparsity of mobility data, it is not an easy task to predict future POIs (place-of-interests) that are going to be visited. In this paper, we propose MobTCast, a Transformer-based context-aware network for mobility prediction. Specifically, we explore the influence of four types of context in mobility prediction: temporal, semantic, social, and geographical contexts. We first design a base mobility feature extractor using the Transformer architecture, which takes both the history POI sequence and the semantic information as input. It handles both the temporal and semantic contexts. Based on the base extractor and the social connections of a user, we employ a self-attention module to model the influence of the social context. Furthermore, unlike existing methods, we introduce a location prediction branch in MobTCast as an auxiliary task to model the geographical context and predict the next location. Intuitively, the geographical distance between the location of the predicted POI and the predicted location from the auxiliary branch should be as close as possible. To reflect this relation, we design a consistency loss to further improve the POI prediction performance. In our experimental results, MobTCast outperforms other state-of-the-art next POI prediction methods. Our approach illustrates the value of including different types of context in next POI prediction.

## 1 Introduction

Human mobility prediction plays an important role in a wide range of applications, including personalised advertisement, traffic modelling, and predicting the spread of pandemics [14, 17]. For example, targeted advertisements, such as store information and coupons can be delivered precisely by predicting the future POIs that users might visit. It could also be valuable in the context of pandemics, to support contact tracing and crowd management.

Unlike other densely acquired trajectory sequence data (such as trajectories extracted from videos [1, 20] and GPS trajectories [21, 29]), trajectory data for the next POI prediction task is typically in the form of check-in sequences and collected from popular Location Based Social Networks (LBSNs), such as Foursquare. As the visited POI information is recorded only when a user chooses to check-in

35th Conference on Neural Information Processing Systems (NeurIPS 2021).

and accesses the location service, the check-in sequence data is very sparse. Due to privacy concerns or battery limitations on mobile devices, users rarely report their locations all the time. This inherent sparsity increases the difficulty of the prediction of future POIs dramatically.

In the last few years, given major advances in deep learning techniques and especially the success of applying Recurrent Neural Networks (RNNs) for sequence prediction tasks, many research works have leveraged the user's historical trajectories to predict most likely POI to be visited by that user. A pioneering work proposed ST-RNNs [25] which employ both time-specific and distance-specific transition matrices in an RNN architecture to model time intervals and geographical distances between two consecutive check-in POIs in historical trajectories. DeepMove [9] incorporates a historical attention module to capture the periodicity of human mobility. Following this trend, more recently, VANext [11] and ACN [27] propose two more attention mechanisms to learn long term preferences for users from history trajectories. However, considering the data sparsity, only exploring the historical visiting information may not be enough. For instance, compared to the large number of POIs and the complexity of the nature of people's preferences, the small amount of visited POIs in a user's historical log is unlikely to reveal their entire range of preferences over all available POIs. Therefore, in addition to exploring the *temporal context* based on the user's own past visiting history, the influence from other types of context could be valuable to achieve more accurate POI prediction. In particular, there are three important types of context that could help tackle the sparsity problem: (1) *Semantic Context*: Typically, in LBSNs, each POI belongs to a category with high level semantic meanings such as *Shops* and *Education*. This context is an indicator in predicting the next POI. For example, at lunchtime, people are more likely to visit Food POIs than other types of POIs. Leisure travellers will tend to check in at tourism POI attractions. (2) *Social Context*: Friends and family members often visit a POI together. A user may also check in a new POI recommended by friends. (3) *Geographical Context*: POIs that are close to each other might be visited by the same user due to their proximity. For example, a user may visit a shopping mall and then go to a nearby cinema. A detailed data-driven analysis of these contexts is given in Section 7.2.

However, one limitation in the existing literature is that the context has been typically modelled based on manually designed functions. For example, Sun *et al.* [30] noticed that the geographical distances between any two non-consecutive visited POIs are also important. A geo-nonlocal function is introduced into the RNN to measure the distance between each visited POI and the average location of all visited POIs. Similarly, Flashback [38] introduces a manually designed weighting function to assign weights based on both temporal and geographical distances for each time step.

In this paper, we propose a novel method called MobTCast, for the next POI prediction task, which is explicitly designed to model and incorporate the influence of the aforementioned contexts in a data-driven manner. A Transformer [31]-based mobility feature extractor is a fundamental component in MobTCastto perform the main POI prediction task. It takes both the visited POI history and the semantic category information to learn semantic-aware mobility features. Using this mobility feature extractor, we extract not only the semantic-aware mobility features of a user but also the features of his/her neighbours. A self-attention mechanism is then applied to model the social influence from neighbours to the user. This aggregated feature is both semantic and social-aware. To include geographical context, the model includes an auxiliary task, which is different from previously proposed methods that depend on manually designed kernels. Considering that the focus of this auxiliary task is exploring the geographical information (i.e., geographical coordinates of visited POIs), it is designed as a trajectory forecasting task. The purpose is to predict a future location (coordinate) based on the previously visited locations.

As a result of these two tasks (main and auxiliary), there are two predicted coordinates available: the predicted location from the auxiliary trajectory forecasting task and the inferred location from the predicted POI given by the main task. Ideally, the Euclidean distance between these two locations should be as close as possible. To this end, a novel consistency loss function is proposed to reflect and embody this relationship in the training process of MobTCast. Thus, the main and the auxiliary tasks are connected not only at the feature level but also at the output level. We conduct extensive experiments on three publicly available real-world datasets. The results demonstrate MobTCastś competitive POI prediction performance and show the value of incorporating different types of context as well as the consistency loss function.

In summary, the main contributions of our work are two-fold: (i) We address the next POI prediction problem by including four types of context: temporal, geographical, social and semantic. Moreover,

we design an auxiliary trajectory forecasting task to predict the geographical coordinates of the next most likely location. To the best of our knowledge, this is the first time that auxiliary trajectory forecasting is incorporated in a POI prediction model, which provides a new perspective for understanding human mobility. (ii) We introduce a novel consistency loss function to connect the predicted POI (from the main task) and the predicted location (from the auxiliary task). This consistency loss function helps further improve the POI prediction performance.

## 2  Related Work

**Next POI Prediction** One of the earliest works that utilised geographical, temporal, and social contexts to predict the next POI is [6], which combines the periodic day-to-day mobility patterns with information from social ties, inferred from the friendship network. Since this seminal work, many others have proposed methods to better model the check-in behaviours or the trajectory of geographical movements, depending on the data they work on. Based on weight matrix factorisation, Cheng *et al.* [4] proposed a personalised Multi-centre Gaussian Model (MGM) to capture the geographical influence and Lian *et al.* [23] proposed GeoMF model that considers the geographical influence. Li *et al.* [22] developed a ranking based geographical factorisation method to incorporate the influences of geographical and temporal context. Recently, a local geographical model is introduced into the logistic matrix factorisation to improve the accuracy of POI recommendation [28]. GERec [32] employs a spatio-temporal relation path embedding to model the temporal, spatial and semantic content for the next POI recommendation.

Deep learning-based neural networks have been used for temporal sequence modelling. RNNs (including LSTM [15] and GRU [7]) and Transformer [31] have also been proposed and successfully applied to many time series prediction problems such as pedestrian trajectory prediction [1, 36] and traffic prediction [34]. For the next POI prediction task, ST-RNN (Spatial Temporal Recurrent Neural Networks), proposed by Liu *et al.* [25], improves the original RNN architecture by incorporating temporal and spatial context at each time interval in the historical trajectory. Following the trend of using RNN, a series of methods are proposed to model human mobility behaviours and predict the next POI, such as DeepMove [9]. Based on RNNs, it applies an attention mechanism to capture the periodical features of human mobility. Similarly, Gao *et al.* [11] introduced a variational attention mechanism to learn the attention on the historical trajectories.

Efforts have also been made to incorporate different kinds of context for POI prediction. SERM [40] is designed to jointly learn embeddings of temporal and semantic contexts (i.e., text messages). Gao *et al.* [13] proposed to leverage a knowledge graph to accommodate the influence of users, locations and semantics. Flashback [38] tackles the geographical context through an explicitly designed spatio-temporal weighting function. The calculated weights are applied to the hidden states in an RNN correspondingly. STAN [26] incorporates the geographical context to learn the regularities between non-adjacent locations and non-contiguous visits. In these methods, the modelling of the geographical context heavily depends on manually designed kernel functions (e.g., Euclidean distance based functions) or a pre-computed spatio-temporal relation matrix [26] (which would dramatically increase the memory and computation costs when the total number of POIs is large). There is limited work that have simultaneously considered temporal, semantic, social and geographical contexts.

**Auxiliary Task Learning** An auxiliary task is a task that is trained jointly with the main task during the training phase to enhance the main task's performance. As a variant of multi-task learning, auxiliary learning has become popular in recent years. It is mainly used in the computer vision field such as semantic segmentation [5, 35].

Recently, several methods have been proposed using multi-task learning to model human mobility. For example, Chen *et al.* [3] introduced DeepJMT which jointly predicts location and time. MCARNN, proposed by Liao *et al.* [24], is a multi-task network to predict both activity and location. Compared to multi-task learning, in auxiliary task learning the auxiliary task is nonoperational during the inference phase. Note that in MCARNN [24], the authors also proposed to leverage an auxiliary task to learn location embeddings. However, the auxiliary task in the proposed MobTCast approach is a trajectory forecasting task, which is designed to consider the geographical context directly, with the introduction of a consistency loss function to connect the main task and the auxiliary task.

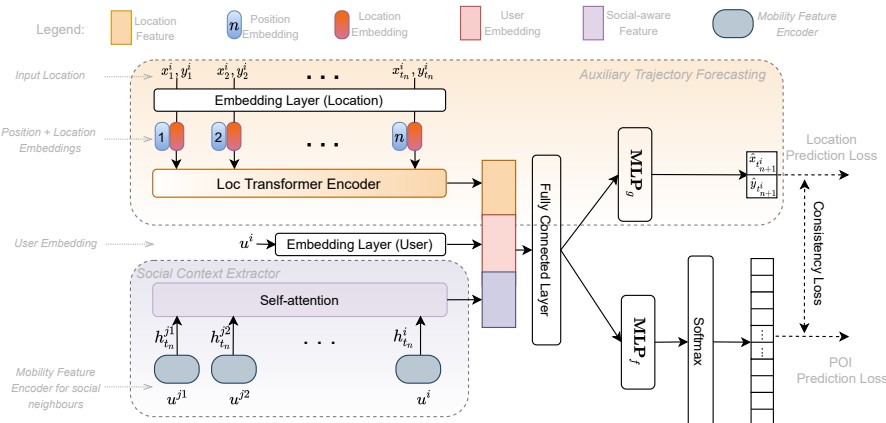

Figure 1: The proposed MobTCast. Dash arrows indicate operations used in the training phase only. Due to space limitations, please refer to Figure 2(a) for details of the Mobility Feature Encoder.

## 3 Problem Formulation

Consider two sets: $\mathbf{P} = \{poi_1, poi_2, \cdots, poi_{|P|}\}$ and $\mathbf{U} = \{u^1, u^2, \cdots, u^{|U|}\}$, denoting the set of POIs and the set of users, respectively. Each POI is represented by a $(lon, lat)$ tuple, which indicates the geographical context. The human mobility prediction problem is defined as follows. Given the history mobility trajectory $T^i = (p_{t_1}^i, p_{t_2}^i, \cdots, p_{t_n}^i)$ of user $u^i \in \mathbf{U}$, the goal is to predict the POI $\hat{p}_{t_{n+1}}^i$ that is going to be visited by the same user at the next time $t_{n+1}^i$. Here, $p_{t_n}^i \in \mathbf{P}$ stands for the POI visited by user $u^i$ at time $t_n$ and the geographical coordinate of $p_{t_n}^i$ is indicated by $(x_{t_n}^i, y_{t_n}^i)$. The observation length of the given history trajectory is $n$.

Based on the geographical coordinates, the semantic category information $c_{t_n}^i$ can be acquired. Note that depending on the dataset, we utilise two types of POI category hierarchies given by: ArcGIS[1] and Foursquare[2]. The social context of user $u^i$, is given by his/her neighbours, represented as $\mathcal{N}_{(i)} = \{u^{j1}, u^{j2}, \cdots, u^{jm}\}$, which is a group of users (in $\mathbf{U}$) socially connected with user $u^i$.

## 4 Method

The overall framework of the proposed method is illustrated in Figure 1. To solve the POI prediction task and the auxiliary trajectory forecasting task, MobTCast consists of four major components: (i) **Mobility Feature Extractor** (Section 4.1): this is a fundamental component in the proposed POI prediction method. It is used to extract semantic-aware mobility features of a given history trajectory. The temporal context is also modelled in this extractor. (ii) **Social Context Extractor** (the light purple part in Figure 1, Section 4.2): this module is used to model the social influence of user $u^i$ in the prediction process. (iii) **Auxiliary Trajectory Forecasting** (the light orange part in Figure 1, Section 4.3): this auxiliary task is introduced to incorporate the geographical context in the POI prediction, to explicitly predict the next geo-location. (iv) **Consistency Loss**: (Section 4.4) this component establishes a relationship between the auxiliary trajectory forecasting and the POI prediction. The detailed description of each component is presented in the following subsections.

### 4.1 Semantic-aware Mobility Feature Extractor

The modelling of the given history trajectory of a user is an essential step in any POI prediction method. In MobTCast, this step is tackled as a mobility feature extraction process. In addition to the visited POIs and the visited timestamps that have been widely considered by other methods [9], we argue that the semantic context plays a significant role and should also be incorporated in this process. For example, *Alice* visited a *Sushi Shop* for lunch yesterday but she decides to go to *McDonald's* today. Although *Sushi Shop* and *McDonald's* are two different POIs, they belong to the same semantic

---

[1] https://developers.arcgis.com/rest/geocode/api-reference/geocoding-category-filtering.htm
[2] https://developer.foursquare.com/docs/build-with-foursquare/categories/

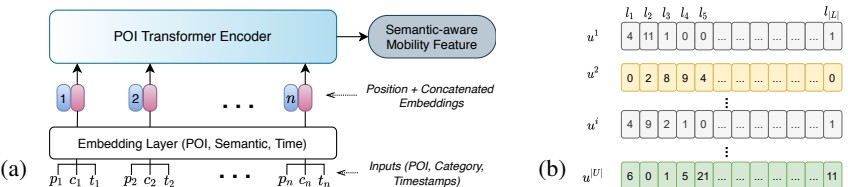

Figure 2: (a) Illustration of the proposed Transformer-based mobility feature extractor. (b) Illustration of using check-in vectors for discovering neighbours ( $u^1$ is a neighbour of $u^i$).

category *Food*. Such a high-level semantic context is helpful to predict which POI is going to be visited by *Alice* tomorrow noontime.

To incorporate the semantic context, a Semantic-aware Mobility Feature Extractor is explicitly designed. As shown in Figure 2(a), this feature extractor takes three types of inputs: POIs, timestamps, and semantic category labels. The mobility feature extraction process can be formulated as:

$$e_p^{i,t} = \phi_p(p_t^i; \mathbf{W}_{\phi_p}), \ e_c^{i,t} = \phi_c(c_t^i; \mathbf{W}_{\phi_c}), \ e_t^{i,t} = \phi_t(t_t^i; \mathbf{W}_{\phi_t}), \tag{1}$$

$$e^{i,t} = e_p^{i,t} \oplus e_c^{i,t} \oplus e_t^{i,t}, \tag{2}$$

$$h_{t_n}^i = \mathcal{F}(e^{i,t_1}, e^{i,t_2}, \cdots, e^{i,t_n}; \mathbf{W}_{\mathcal{F}}), \tag{3}$$

where $\oplus$ is the concatenation operation and $e^{i,t}$ is the concatenated embedding of: the POI embedding $e_p^{i,t}$, the category embedding $e_c^{i,t}$, and the temporal embedding $e_t^{i,t}$. In these equations, the $\mathbf{W}$ terms stand for the weight matrices that need to be optimised during the training process. These three embedding vectors are calculated through three different embedding layers ($\phi_p(\cdot)$, $\phi_c(\cdot)$, and $\phi_t(\cdot)$), respectively. In Eq. (3), $\mathcal{F}(\cdot)$ is a function to extract feature $h_{t_n}^i$ with embedding vectors of the whole history trajectory from $t_1$ to $t_n$ as input. This extracted feature $h_{t_n}^i$ models the spatio-temporal history movement behaviour of user $u^i$ and it is semantic-aware as well. In MobTCast, the Transformer architecture [31] is used as $\mathcal{F}(\cdot)$ as it has been proven effective in other sequence modelling tasks, such as language understanding [8], speech recognition [33], and flow prediction [37]. The Transformer architecture is based on a self-attention mechanism so it can better model the history trajectory movement than a vanilla RNN structure. Since there is no recurrence or convolution operation in the Transformer, position embeddings are added to the concatenated embeddings to retain sequential order information (see Figure 2(a)). We use standard sine/cosine function based position embeddings introduced in [31].

## 4.2 Social Context Extractor

The social influence is another important kind of context to be considered in the next POI prediction process, especially when the mobility data is sparse. For example, one of *Alice*'s friends recommends a new restaurant that she has never visited before, so she decides to visit it tomorrow. In this example, since there is a high likelihood that this restaurant has been visited by *Alice*'s friend before, the social context (*Alice*'s friend's mobility) is helpful to predict *Alice*'s next POI.

Inferring social links based on mobility data is yet another challenging task [2, 10, 16]. Since our main focus is POI prediction, we adopt a straightforward approach to discover the neighbour set $\mathcal{N}_{(i)}$ of user $u^i$. The social neighbour discovering process is as follows. As shown in Figure 2(b), we first build a check-in vector for each user, based on their check-in history. The dimension of this check-in vector equals the number of POIs ($|P|$) and each dimension of the vector indicates how many times the user visited the corresponding POI. The cosine similarity of two users' check-in vectors is then computed. If the calculated cosine distance is larger than a threshold $\tau$, these two users are considered to be connected.

For each user $u^j \in \mathcal{N}_{(i)}$, the mobility feature extractor (Eqs. (1) - (3)) is applied to calculate the semantic-aware mobility feature $h_{t_n}^j$. It worth noting that any "future" information (after $t_n^i$) from neighbours cannot be used as it is not available for the prediction time step $t_{n+1}^i$. If $t_n^j$ is the last timestamp of a neighbour's observed trajectory, mathematically, the above condition can be represented as: $t_n^j \leq t_n^i, \forall j \in \mathcal{N}_{(i)}$. The extraction of social influence can be seen as a feature aggregation process. The social context vector $H_{t_n}^i$ is a weighted sum of his/her own mobility feature

$h_{t_n}^i$ and his/her neighbour's (all neighbours in $\mathcal{N}_{(i)}$ are considered) mobility feature $h_{t_n}^j$:

$$H_{t_n}^i = \alpha_i h_{t_n}^i + \sum_{j \in \mathcal{N}_{(i)}} \alpha_j h_{t_n}^j \tag{4}$$

To compute the weight terms $\alpha$, we use a multi-head self-attention mechanism [31].

## 4.3 Auxiliary Trajectory Forecasting

Compared to other recommendation tasks, like next shopping item prediction and music recommendation, the geographical proximity of POIs is an inherent characteristic in the next POI prediction task. For example, *Alice* is more likely to visit POIs (e.g., a *supermarket* to buy snacks or a *park* to walk) that are geographically close to the visited *Sushi Shop* than those that are significantly further away. This geographical information is thus another important type of context that is worthy to exploit in the next POI prediction task.

Unlike previously published efforts [23, 41, 13, 38] that take the geographical information into account by manually designed kernel functions, the geographical context is incorporated in MobTCast by leveraging an auxiliary task. This auxiliary task is designed as a trajectory forecasting task. Since POI prediction is typically tackled as a classification task, the long tail distribution of POIs caused by data sparsity is a severe bottleneck that limits the performance. Thus, the motivation of this introduced auxiliary trajectory forecasting is to directly forecast geo-location coordinates (as a location regression task) to leverage the geographical context and alleviate the effect of data sparsity.

Similar to the pedestrian trajectory prediction task [19, 36], the coordinates of the historical trajectory are firstly embedded through an embedding layer $\phi_l$. Then, another Transformer $\mathcal{F}_g(\cdot)$ is used to extract the geographical context vector $g_{t_n}^i$. Mathematically, this process is expressed as follows:

$$e_l^{i,t} = \phi_l(x_t^i, y_t^i; \mathbf{W}_{\phi_l}), \tag{5}$$

$$g_{t_n}^i = \mathcal{F}_g(e_l^{i,t_1}, e_l^{i,t_2}, \cdots, e_l^{i,t_{n-1}}; \mathbf{W}_{\mathcal{F}_g}), \tag{6}$$

where $\mathbf{W}$ terms are trainable weights. To build a connection between the main task and the auxiliary task, the social context vector $H_{t_n}^i$ from the social context extractor (Section 4.2) and the geographical context vector $g_{t_n}^i$ are concatenated and then passed through a fully connected layer (FC($\cdot$)). In addition, the user embedding that learns the personalised user preference is also included as the user's personal preference would also influence his/her own mobility pattern:

$$e_u^i = \phi_u(u^i; \mathbf{W}_{\phi_l}), \tag{7}$$

$$f_{t_n}^i = \text{FC}(h_{t_n}^i \oplus g_{t_n}^i \oplus e_u^i). \tag{8}$$

Since the user embedding $e_u^i$ remains the same during the whole trajectory, it is only used in Eq. (8) and not used at every time step like other embeddings in Eq. (3) or Eq. (6).

This aggregated feature representation $f_{t_n}^i$ inherits the semantic context, the social context, the geographical context, and the user personal preference. It then can be used for both the main (i.e. predicting the next POI, detailed in the following subsection) and the auxiliary tasks. The predicted location $(\hat{x}_{t_{n+1}^i}, \hat{y}_{t_{n+1}^i})$ at time step $t_{n+1}^i$ is then decoded via:

$$(\hat{x}_{t_{n+1}^i}, \hat{y}_{t_{n+1}^i}) = \text{MLP}_g(f_{t_n}^i), \tag{9}$$

where $\text{MLP}_g(\cdot)$ is a multi-layer perceptrons (MLP)-based decoder.

## 4.4 Next POI Prediction

Based on the context-aware feature $f_{t_n}^i$, the prediction of POI at next time step $t_{n+1}^i$ is as follows:

$$\mathbf{Pr}(\hat{p}_{t_{n+1}}^i) = \text{softmax}(\text{MLP}_f(f_{t_n}^i)), \tag{10}$$

where $\text{MLP}_f(\cdot)$ is another MLP structure. Different from the auxiliary trajectory forecasting (Eq. (9)), an extra softmax layer is required for generating the probabilities of all POIs. Thus, by performing Eq. (10), the most likely POI that will be visited by the user $u^i$ at the time step $t_{n+1}^i$ is the one with the largest probability in $\mathbf{Pr}(\hat{p}_{t_{n+1}}^i)$.

In addition to the POI prediction loss $\mathcal{L}_1$ for the main task and the trajectory forecasting loss $\mathcal{L}_2$ for the auxiliary task, we propose a novel consistency loss $\mathcal{L}_3$ for training MobTCast. The details of these three losses are described below.

**POI Prediction Loss** Given that the next POI prediction is treated as a classification over the entire POI set, the POI prediction loss is a multi-class cross entropy loss:

$$\mathcal{L}_1 = -\sum_{p=1}^{|P|} \mathbf{Pr}(p_{t_{n+1}}^i)_p \log(\mathbf{Pr}(\hat{p}_{t_{n+1}}^i)_p), \tag{11}$$

where $\mathbf{Pr}(p_{t_{n+1}}^i)_p$ indicates the ground truth, i.e., $\mathbf{Pr}(p_{t_{n+1}}^i)_p = 1$ if $u^i$ visited the $p$-th POI at $t_{n+1}^i$. Similarly, $\mathbf{Pr}(\hat{p}_{t_{n+1}}^i)_p$ is the predicted probability that the user $u^i$ will visit the $p$-th POI.

**Trajectory Forecasting Loss** This loss is used for the auxiliary task. It is a mean square error loss that measures the difference between the predicted location and the ground truth location.

$$\mathcal{L}_2 = \|\hat{x}_{t_{n+1}^i} - x_{t_{n+1}^i}\|^2 + \|\hat{y}_{t_{n+1}^i} - y_{t_{n+1}^i}\|^2, \tag{12}$$

where $(x_{t_{n+1}^i}, y_{t_{n+1}^i})$ is the geographical coordinate of the ground truth POI $p_{t_{n+1}^i}$ that is visited by $u^i$ at $t_{n+1}^i$.

**Consistency Loss** To better leverage the auxiliary task, we not only connect the main task and the auxiliary task at the feature level (Eq. (8)) but also design a new loss function to further link the two tasks. Intuitively, there should be a relationship between the predicted POI from the main task and the predicted location from the auxiliary task. That is, the location of the predicted POI and the predicted location should be as close as possible. Consequently, a consistency loss is introduced to embody the above relationship.

Based on the probabilities $\mathbf{Pr}(\hat{p}_{t_{n+1}}^i)$ given by Eq. (10), the location of the predicted POI can be inferred through:

$$\text{argmax}(\mathbf{Pr}(\hat{p}_{t_{n+1}}^i)) \Rightarrow (\tilde{x}_{t_{n+1}^i}, \tilde{y}_{t_{n+1}^i}). \tag{13}$$

The consistency loss is then defined as the mean squared error between the inferred location $(\tilde{x}_{t_{n+1}^i}, \tilde{y}_{t_{n+1}^i})$ and the predicted location $(\hat{x}_{t_{n+1}^i}, \hat{y}_{t_{n+1}^i})$:

$$\mathcal{L}_3 = \|\tilde{x}_{t_{n+1}^i} - \hat{x}_{t_{n+1}^i}\|^2 + \|\tilde{y}_{t_{n+1}^i} - \hat{y}_{t_{n+1}^i}\|^2. \tag{14}$$

Note that, as given in Eq. (15), both $\mathcal{L}_2$ and $\mathcal{L}_3$ (mean mean squared errors) are calculated and averaged over all training samples in a batch.

The overall loss function is a combination of the above three losses so that the parameters of our MobTCast are optimised by minimising:

$$\mathcal{L} = \sum_{b=1}^{B} (\theta_1 \mathcal{L}_1^b + \theta_2 \mathcal{L}_2^b + \theta_3 \mathcal{L}_3^b), \tag{15}$$

where $\theta$ terms are used to balance the three losses and $B$ is the total number of training samples. For the $b$-th training sample, $\mathcal{L}_1^b$, $\mathcal{L}_2^b$, and $\mathcal{L}_3^b$ are the POI prediction loss, trajectory forecasting loss, and the consistency loss, respectively. The coordinates of each location are normalised so that the scale of three losses are similar. Using this combined loss function, MobTCast is trained together in an end-to-end fashion.

## 5 Experiments

In our experiments, we use three widely used LBSNs datasets: Gowalla [6], Foursquare-NYC [39] (FS-NYC), and Foursquare-Tokyo [39] (FS-TKY) (more details of these datasets are contained in Section 7.1 of the *Appendix*). These datasets are publicly available and no personally identifiable information is included. Considering the social connections for FS-NYC and FS-TKY are not given, the method described in Section 4.2 (with $\tau = 0.5$) is used to explore the neighbours of each user. In our experiments, all visiting records in the training set are used to build check-in vectors so that the social connection of any two users is fixed. Following [38], the observation length $n$ is set to 20 and we split the check-in sequence of each user into 80% for training and 20% for testing.

Table 1: Performance results (mean $\pm$ std ) of different methods on the three datasets. In each row, the best performing results are shown in bold and the second best are given in underline.

| | Acc | FMFMGM | LGLMF | RNN | DeepMove | Flashback | STAN | MobTCast |
|---|---|---|---|---|---|---|---|---|
| Gowalla | Top-1 | 0.0394(0.0035) | 0.0605(0.0019) | 0.1334(0.0021) | 0.1636(0.0019) | 0.1809(0.0023) | 0.1729(0.0015) | **0.2051(0.0022)** |
| | Top-5 | 0.1218(0.0050) | 0.1842(0.0049) | 0.3040(0.0017) | 0.3704(0.0021) | 0.3918(0.0018) | 0.3875(0.0033) | **0.4364(0.0015)** |
| | Top-10 | 0.1930(0.0058) | 0.2793(0.0062) | 0.3717(0.0025) | 0.4536(0.0022) | 0.4710(0.0021) | 0.4816(0.0038) | **0.5236(0.0022)** |
| | Top-20 | 0.2976(0.0029) | 0.3912(0.0050) | 0.4315(0.0026) | 0.5196(0.0031) | 0.5372(0.0014) | 0.5534(0.0028) | **0.5956(0.0020)** |
| FS-NYC | Top-1 | 0.1201(0.0011) | 0.0591(0.0038) | 0.1960(0.0025) | 0.2517(0.0021) | 0.2602(0.0028) | 0.2755(0.0036) | **0.2804(0.0024)** |
| | Top-5 | 0.3103(0.0014) | 0.2122(0.0069) | 0.5258(0.0063) | 0.5929(0.0032) | 0.5992(0.0028) | 0.6089(0.0033) | **0.6591(0.0031)** |
| | Top-10 | 0.4054(0.0023) | 0.3248(0.0089) | 0.6535(0.0102) | 0.7013(0.0041) | 0.7192(0.0054) | 0.7427(0.0037) | **0.7816(0.0047)** |
| | Top-20 | 0.4967(0.0009) | 0.4520(0.0113) | 0.7464(0.0058) | 0.7763(0.0045) | 0.8079(0.0038) | 0.8398(0.0033) | **0.8561(0.0041)** |
| FS-TKY | Top-1 | 0.0234(0.0061) | 0.0334(0.0064) | 0.1775(0.0017) | 0.1927(0.0027) | 0.2303(0.0020) | 0.2238(0.0035) | **0.2550(0.0048)** |
| | Top-5 | 0.0690(0.0128) | 0.1271(0.0209) | 0.4389(0.0026) | 0.5023(0.0022) | 0.5331(0.0027) | 0.5293(0.0039) | **0.5683(0.0055)** |
| | Top-10 | 0.1408(0.0227) | 0.2007(0.0110) | 0.5397(0.0047) | 0.5909(0.0049) | 0.6346(0.0054) | 0.6245(0.0058) | **0.6726(0.0042)** |
| | Top-20 | 0.2174(0.0178) | 0.3495(0.0147) | 0.6211(0.0062) | 0.6720(0.0065) | 0.7082(0.0051) | 0.7134(0.0047) | **0.7489(0.0054)** |

The hyperparameters are set based on the performance on the validation set which is 10% of the training set. The hidden dimensions for the Transformer used in the mobility feature extractor and the auxiliary task are both 128. According to Eq. (2) and (3), the sum of the dimensions of the POI, semantic and temporal embeddings equals the hidden dimensions of the Transformer $\mathcal{F}(\cdot)$. Thus, we set the dimensions of these embeddings as 80, 24, and 24. As for the weights of three loss functions in Eq. (15), all three $\theta$s are set to 1. For the auxiliary task, we process the dataset by normalising the coordinates (latitude and longitude) within [-1, 1]. The model is trained using an Adam optimiser [18] and implemented using PyTorch on a desktop with an NVIDIA GeForce RTX-2080 Ti GPU.

To evaluate the performance of the proposed method, we compare MobTCast against the following POI prediction approaches: FMFMGM [4], a fused matrix factorisation framework using the Multi-centre Gaussian Model to model the geographical information. LGLMF [28], a logistic matrix factorisation method combined with a local geographical model for leveraging geographical information. RNN, the basic architecture for sequential predictions. In the experiments, we use the GRU variant [7] as the baseline. DeepMove [9], a popular RNN-based POI prediction model that incorporates an attention mechanism for modelling the POI visiting history. Flashback [38] and STAN [26], two recent state-of-the-art deep learning-based models for next POI prediction. Similar to [9], we use top-$k$ accuracy ($k = 1, 5, 10, 20$) to evaluate the POI prediction performance. The predicted probabilities of all POIs ($\mathbf{Pr}(\hat{p}_{t_{n+1}}^i)$ yield from the main task) are ranked from large to small. We then check whether the actual visited POI ($p_{t_{n+1}}^i$) appears in the top-$k$ predicted POIs.

## 5.1 Prediction Performance

The results of MobTCast and the baseline methods are given in Table 1. The best performing method on each dataset is given in bold and the second best in underline. For each model, we ran the experiments 5 times and report the average accuracy as well as the standard deviation. In general, deep learning-based methods outperform matrix factorisation-based methods (FMFMGM and LGLMF) by a relatively large margin. The baseline RNN has the worst performance among all deep learning-based methods as no attention mechanism or any context is taken into account. Compared to DeepMove, the weighting function proposed in Flashback is based on temporal and geographical distances, which results in better prediction performance. Overall, with more contexts and the auxiliary task, MobTCast achieves the best performance in comparison to other methods across all datasets.

The original social links of users may be not available in the general case, due to e.g., privacy concerns. The experiments on the FS-NYC and FS-TKY datasets, where social links are not provided, demonstrate that MobTCast is able to handle these datasets by applying a visiting pattern-based process (described in Section 4.2) to discover the "social" context for POI prediction. Such discovered social context can provide useful information for prediction. This also shows the generalisation of MobTCast on different datasets. The average improvement of MobTCast (across all three datasets and all four accuracy metrics) is 7.22%. This improvement is computed as $(acc_1 - acc_2)/acc_2 \times 100\%$, where $acc_1$ and $acc_2$ are the accuracy of MobTCast and the second best performing method (either Flashback or STAN depending on the metrics and datasets). These results illustrate the superior prediction performance of the proposed method.

Table 2: The prediction results of the ablation studies on all three datasets, without (w/o) the specifically mentioned components.

| | Gowalla | | | | FS-NYC | | | | FS-TKY | | | |
|---|---|---|---|---|---|---|---|---|---|---|---|---|
| | TOP-1 | Top-5 | Top-10 | Top-20 | TOP-1 | Top-5 | Top-10 | Top-20 | TOP-1 | Top-5 | Top-10 | Top-20 |
| V0 (w/o all contexts) | 0.1432 | 0.3374 | 0.4129 | 0.4753 | 0.2219 | 0.5628 | 0.6921 | 0.7723 | 0.1793 | 0.4898 | 0.5882 | 0.6612 |
| V1 (w/o geographical, w/o social) | 0.1691 | 0.3747 | 0.4528 | 0.5192 | 0.2466 | 0.6062 | 0.7338 | 0.8045 | 0.2018 | 0.4966 | 0.5984 | 0.6739 |
| V2 (w/o social, w/o $\mathcal{L}_2$, w/o $\mathcal{L}_3$) | 0.1896 | 0.4210 | 0.5058 | 0.5766 | 0.2504 | 0.6133 | 0.7417 | 0.8269 | 0.2367 | 0.5459 | 0.6504 | 0.7271 |
| V3 (w/o social, w/o $\mathcal{L}_3$) | 0.1925 | 0.4231 | 0.5097 | 0.5796 | 0.2622 | 0.6151 | 0.7476 | 0.8328 | 0.2395 | 0.5463 | 0.6537 | 0.7304 |
| V4 (w/o social, w/o $\mathcal{L}_2$) | 0.1934 | 0.4242 | 0.5098 | 0.5813 | 0.2635 | 0.6293 | 0.7606 | 0.8363 | 0.2407 | 0.5483 | 0.6535 | 0.7299 |
| V5 (w/o social) | 0.1987 | 0.4332 | 0.5203 | 0.5948 | 0.2727 | 0.6455 | 0.7649 | 0.8457 | 0.2469 | 0.5571 | 0.6641 | 0.7406 |
| MobTCast | **0.2051** | **0.4364** | **0.5236** | **0.5956** | **0.2804** | **0.6591** | **0.7816** | **0.8561** | **0.2550** | **0.5683** | **0.6726** | **0.7489** |

## 5.2 Ablation study

To investigate the effectiveness of each module in our proposed method, we consider 6 different variants in total (detailed configurations of these variants are given in Section 7.3). The first two variants are: (i) **V0**: None of the three contexts is incorporated in this variant. It can be seen as a mobility feature extractor without using semantic information as input. (ii) **V1**: This variant includes the semantic context only. It predicts the next POI based on the semantic-aware mobility features. Furthermore, given that incorporating the geographical context through the auxiliary task introduces two more loss functions (i.e., trajectory forecasting loss and consistency loss), we also explore the contribution of each loss function through the following 4 variants: (iii) **V2**: This variant only uses the basic POI prediction loss from the main task. (iv) **V3**: The loss function for this variant is a combination of the POI prediction loss and the trajectory forecasting loss from the auxiliary task. (v) **V4**: For this variant, it employs both the POI prediction loss and the proposed consistency loss. (vi) **V5**: All three loss functions are combined in this variant. This loss function is equivalent to the loss function used in MobTCast. For V2 - V4, both semantic and geographical context are enabled. However, the social context extractor used in MobTCast to model the social context is replaced with the basic semantic-aware feature extractor, which means the social context is excluded.

The results (average of 5 runnings) of these variants on the three datasets are given in Table 2. In general, comparing the results from MobTCast against all six variants, it can be noticed that MobTCast outperforms all the variants on all three datasets. V0 yields the worst performance, which is as expected as it is a basic prediction model based on the visiting history without considering any contextual information. When the semantic context is incorporated in V1, compared to V0, we observe that there is a performance gain on all three datasets. With the help of the auxiliary task, variants V2 - V5 have better performances than V0 and V1. More precisely, among these four variants, the performance of V2 is worse than that of the other variants as only the POI prediction loss from the main task is applied and there are no extra constraints for the auxiliary task in the training process. On the contrary, V5 achieves the top performance, where all three loss functions are combined. When only two loss functions are used, V4 outperforms V3 by a relatively small margin. These comparison results confirm the importance of combining both the trajectory forecasting loss and consistency loss when the auxiliary task is introduced. Especially for the consistency loss, it can work as a connection between the two tasks and further improve the POI prediction performance. The results of the above ablation studies show the effectiveness of each module and justify the need for all loss functions included in MobTCast.

## 5.3 Limitations and Future Work

A limitation of our approach lies in that MobTCast depends on contextual information, such as semantic categories. If semantic contexts of places (or social connections) are changed over time, the performance of a trained model might be not optimal. Also, given that the social context extractor takes every neighbour into consideration, the efficiency of our MobTCast would drop if there are too many neighbours. We plan to investigate applying graph neural network structures such as message passing neural networks [12] and simple spectral graph convolution [42] to model the social context in future work. We plan to address these two limitations in future work.

## 6 Conclusion

In this paper, we have proposed MobTCast, a novel deep learning-based architecture to tackle the next POI prediction task. It contains a Transformer-based semantic-aware mobility feature extractor, and a

social context extractor to consider the mobility features of the user's neighbours to take into account the social influence. Furthermore, it includes an auxiliary task, which is a trajectory forecasting task, to incorporate the geographical context. Through extensive experiments, the results illustrate that MobTCast outperforms other state-of-the-art prediction methods and demonstrate the effectiveness of incorporating each type of context in MobTCast. In addition, the main task and the auxiliary task are connected via a consistency loss function. The ablation study results show that introducing the consistency loss function can further improve the performance.

## Acknowledgments and Disclosure of Funding

This research is supported by Australian Research Council (ARC) Discovery Project *DP190101485*.

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
