# 7  Appendix

## 7.1  Details of Datasets

The details of datasets used in this work are summarised in Table 3. For each dataset, to acquire the user's check-in sequence, the check-in records of each user are sorted chronologically based on the timestamp given in the dataset. In these datasets, there is no personally identifiable information. In addition, noting that in real-world settings, privacy-preservation in human mobility prediction could be achieved by existing strategies such as anonymisation, differential privacy, and also privacy-aware federated learning. For the semantic information, Foursquare dataset [39] includes low-level semantic category labels (such as Japanese restaurant and Italian restaurant). We then follow the standard Foursquare POI category hierarchy and map each POI to one of the following 8 high-level semantic categories: Arts & Entertainment, College & University, Food, Professional & Other Places, Nightlife Spot, Outdoors & Recreation, Shop & Service, Travel & Transport and Residence. As listed in the table, the semantic category information is not directly given for the Gowalla dataset [6]. ArcGIS is used to decode the semantic information based on the geographical coordinate of each POI. The ArcGIS has the following 11 high-level categories: Arts & Entertainment, Education, Water Features, Travel & Transport, Shops & Service, Residence, Professional & Other places, Parks & Outdoors, Nightlife Spot, Land Features, Food.

In the case of the Gowalla dataset, the included POIs are across different cities, whereas FS-NYC/FS-TKY are only about a single city. As compared in Table 1, the proposed MobTCast works well on all three datasets (relatively large Gowalla and small FS-TKY/FS-NYC), which indicates that the proposed method is robust enough to work well on both large and small region scales. In addition, social links are not provided for FS-NYC and FS-TKY. More details about how to discover the social contexts for these datasets are given in Section 4.2. Experiments on such datasets (semantic or social information is not given) could evaluate the generalisation of our method at a certain level.

Table 3: Details of each dataset.

|  | Gowalla | FS-NYC | FS-TKY |
|---|---|---|---|
| # User | 107,092 | 1,083 | 2,293 |
| # POI | 1,280,969 | 38,334 | 61,859 |
| # Check-in | 6,442,892 | 227,428 | 573,703 |
| Collection Start | 2009/02 | 2012/04 | 2012/04 |
| Collection End | 2010/10 | 2013/02 | 2013/02 |
| Semantic Category | No | Yes | Yes |
| Social Links | Yes | No | No |

## 7.2  Data Driven Analysis

In this part, as a supplement to the discussion in Section 1, we discuss the motivation of incorporating different contexts in POI prediction system from a data-driven perspective.

1. *Semantic Context*. Typically, in LBSNs, each POI belongs to a category with high level semantic meanings such as *Shop* and *Education*. For example, as illustrated in Figure 3(a), there are three visiting peaks (corresponding to breakfast, lunch, and dinner time) for *Food* POIs in each day. As for the *Education*, there is only one morning peak. This shows that POIs with different semantic meanings have different visiting patterns.

2. *Social Context*. Friends and family members often visit a POI together. A user may also check-in a new POI recommended by friends. As shown in Figure 3(b), the average DTW (Dynamic Time Warping, calculated with longitude and latitude coordinates) distance from a user's trajectory to his/her friends' trajectories is much smaller than that to other strangers' trajectories, which reveals that social influence is an important context for predicting POIs.

3. *Geographical Context*. POIs that are close to each other might be visited by the same user due to their proximity. To support and validate the importance of this context, we split each user's trajectory in a short period (e.g., two consecutive days) into multiple trajectory clips based on the temporal interval between two visits. If the interval is larger than 6 hours, the trajectory will be cut into two clips. We then calculate and compare two statistics: i)

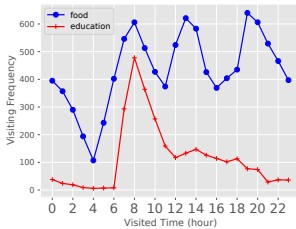
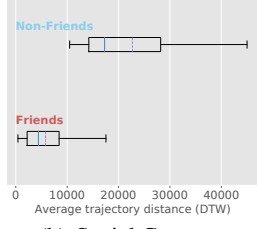
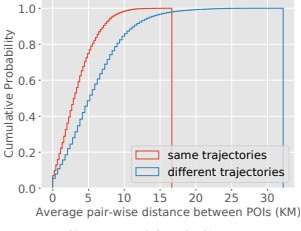

| (a) Semantic Context | (b) Social Context | (c) Geographical Context |

Figure 3: Data-driven motivation of incorporating different contexts. These statistical analysis are based on FS-NYC [39], Gowalla [6], and FS-TKY [39] datasets, respectively.

Table 4: The configuration of each variant.

|          | Semantic | Geographical | Social | Loss |
|----------|----------|--------------|--------|------|
| V0       | $\times$ | $\times$     | $\times$ | $\mathcal{L}_1$ |
| V1       | $\checkmark$ | $\times$ | $\times$ | $\mathcal{L}_1$ |
| V2       | $\checkmark$ | $\checkmark$ | $\times$ | $\mathcal{L}_1$ |
| V3       | $\checkmark$ | $\checkmark$ | $\times$ | $\mathcal{L}_1 + \mathcal{L}_2$ |
| V4       | $\checkmark$ | $\checkmark$ | $\times$ | $\mathcal{L}_1 + \mathcal{L}_3$ |
| V5       | $\checkmark$ | $\checkmark$ | $\times$ | $\mathcal{L}_1 + \mathcal{L}_2 + \mathcal{L}_3$ |
| MobTCast | $\checkmark$ | $\checkmark$ | $\checkmark$ | $\mathcal{L}_1 + \mathcal{L}_2 + \mathcal{L}_3$ |

the average pair-wise distance between POIs in the same trajectories; and ii) the average pair-wise distance between POIs in different trajectories. As it is shown in Figure 3(c), the distributions of these two distances are different and the distance between POIs in the same trajectories is smaller, which justifies the geographical context.

## 7.3 Detailed Configurations of Variants in Ablation Study

All variants compared in Ablation Study (Section 5.2) are summarised in Table 4. In the table, a $\checkmark$ indicates that the corresponding context is incorporated, whereas a $\times$ means that the context is disabled.

## 7.4 Auxiliary Task for POI Prediction

In this part of the experiments, we investigate whether we can use the auxiliary task solely to achieve a good POI prediction performance. We aim to answer the research question: can we only use the auxiliary task to predict next POI? Since the outputs from the trajectory forecasting auxiliary task are predicted coordinates (Eq. (9)) instead of probabilities of POIs, it cannot be directly used for predicting the next POI. A tweak is required to get predicted POI from the predicted location. By calculating the Euclidean distances between the predicted location and the locations of all $|P|$ POIs, a $\mathbb{R}^{|P|}$ vector is obtained. The index of the minimum value in this vector is then considered as the predicted POI.

We design two baselines with the tweak operation:

- Aux-RNN: This baseline is an RNN based trajectory forecasting auxiliary task. To be more specific, the forecasting function $\mathcal{F}_g(\cdot)$ in Eq. (6) is a GRU.

- Aux-Tra: This baseline is the same auxiliary task (Transformer based) used in MobTCast (details are given in Section 4.3).

For both Aux-RNN and Aux-Transformer, the user embedding is also applied.

Table 5 lists the POI prediction performance of Aux-RNN and Aux-Transformer on three datasets. For the convenience of comparison, the performance of the proposed MobTCast is also listed in the last row of the table. As shown in the table, it is very clear that both Aux-RNN and Aux-Transformer have

Table 5: The results of Aux-RNN and Aux-Transformer, where only the auxiliary task is used for POI prediction.

| | Gowalla | | | | FS-NYC | | | | FS-TKY | | | |
|---|---|---|---|---|---|---|---|---|---|---|---|---|
| | TOP-1 | Top-5 | Top-10 | Top-20 | TOP-1 | Top-5 | Top-10 | Top-20 | TOP-1 | Top-5 | Top-10 | Top-20 |
| Aux-RNN | 0.00003 | 0.0002 | 0.0003 | 0.0007 | 0.0011 | 0.0023 | 0.0045 | 0.0092 | 0.0004 | 0.0006 | 0.0014 | 0.0046 |
| Aux-Tra | 0.00003 | 0.0002 | 0.0004 | 0.0008 | 0.0017 | 0.0027 | 0.0051 | 0.0106 | 0.0004 | 0.0007 | 0.0019 | 0.0049 |
| MobTCast | **0.2051** | **0.4364** | **0.5236** | **0.5956** | **0.2804** | **0.6591** | **0.7816** | **0.8561** | **0.2550** | **0.5683** | **0.6726** | **0.7489** |

poor performance on all datasets, which reveals that these auxiliary tasks and the tweak operation cannot fully model the complex check-in behaviour of each user. More specifically, Aux-Transformer is slightly better than Aux-RNN. Comparing their performances across different datasets, it can be seen that they perform better if the total number of POIs is smaller. This correlation is as expected. Overall, these results indicate that only using the auxiliary task cannot yield good POI predictions.

## 7.5 Computational Cost

In this section, we explore the computation cost of our MobTCast. In Table 6, the inference time (running one training/testing instance) of each deep learning-based model is listed. All these recorded times are executed on the same GPU. Among these methods, the basic RNN (the first row) is the fastest as no other context or attention mechanism is included. Compared to RNN, DeepMove is slightly slower due to the introduced attention calculation in the input history trajectory. Because the attention in Flashback depends on the calculation of temporal and geographical distances (using geographical coordinates), Flashback takes longer for running. As STAN incorporates a bi-layer attention architecture (one attention layer for considering spatio-temporal correlation within user trajectory and one attention layer for recalling the most plausible candidates from all POIs), the inference time of STAN is the largest in the table.

As for our MobTCast_V0, it requires more time than RNN and DeepMove (but less than Flashback) for running. This is because that the Transformer structure which adopts the multi-head attention mechanism is used in MobTCast as the backbone. If we compare V1 against V0, it can be seen that incorporating the semantic context is efficient enough (with a marginal cost increment only). For V2-V5, since the auxiliary trajectory forecasting is introduced, there is a notable increase. However, compared to Flashback where the geographical context is also included, the over-cost of each variant is relatively small. We can also notice that the inference times of these four variants are almost the same. The differences between these variants are about the loss functions (i.e., whether including trajectory forecasting loss and consistency loss), which results in very similar computation costs. Although MobTCast (the last row) needs longer running time than other variants, it takes the social context into accounts. Thus, considering that MobTCast incorporates the computation of social neighbours, the increment of inference time is still acceptable.

Table 6: Comparison of computational cost. Each method is benchmarked on the same NVIDIA GeForce RTX-2080 Ti GPU.

| Method | Inference Time ($10^{-6}$ Seconds) |
|---|---|
| RNN | 2.457 |
| DeepMove | 3.095 |
| Flashback | 29.764 |
| STAN | 643.779 |
| MobTCast_V0 | 16.114 |
| MobTCast_V1 | 16.550 |
| MobTCast_V2 | 33.939 |
| MobTCast_V3 | 33.931 |
| MobTCast_V4 | 33.913 |
| MobTCast_V5 | 33.946 |
| MobTCast | 127.242 |

Table 7: Results (on the validation set) of different $\theta_2$ and $\theta_3$ combinations.

| | | $\theta_2 = 0.5$ | $\theta_2 = 1.0$ | $\theta_2 = 2.5$ | $\theta_2 = 5.0$ |
|---|---|---|---|---|---|
| Top-1 Accuracy | $\theta_3 = 0.5$ | 0.2874 | 0.2874 | 0.2910 | 0.2793 |
| | $\theta_3 = 1.0$ | 0.2883 | **0.2919** | 0.2892 | 0.2739 |
| | $\theta_3 = 2.5$ | 0.2811 | 0.2860 | 0.2829 | 0.2721 |
| | $\theta_3 = 5.0$ | 0.2775 | 0.2865 | 0.2789 | 0.2766 |
| | | $\theta_2 = 0.5$ | $\theta_2 = 1.0$ | $\theta_2 = 2.5$ | $\theta_2 = 5.0$ |
| Top-5 Accuracy | $\theta_3 = 0.5$ | 0.6507 | 0.6529 | 0.6538 | 0.6471 |
| | $\theta_3 = 1.0$ | 0.6430 | **0.6560** | 0.6439 | 0.6484 |
| | $\theta_3 = 2.5$ | 0.6408 | 0.6462 | 0.6372 | 0.6426 |
| | $\theta_3 = 5.0$ | 0.6327 | 0.6453 | 0.6444 | 0.6515 |
| | | $\theta_2 = 0.5$ | $\theta_2 = 1.0$ | $\theta_2 = 2.5$ | $\theta_2 = 5.0$ |
| Top-10 Accuracy | $\theta_3 = 0.5$ | **0.8029** | 0.7881 | 0.7925 | 0.7813 |
| | $\theta_3 = 1.0$ | 0.7867 | 0.7921 | 0.7957 | 0.7966 |
| | $\theta_3 = 2.5$ | 0.7809 | 0.7975 | 0.7777 | 0.7930 |
| | $\theta_3 = 5.0$ | 0.7854 | 0.7952 | 0.7836 | 0.7876 |
| | | $\theta_2 = 0.5$ | $\theta_2 = 1.0$ | $\theta_2 = 2.5$ | $\theta_2 = 5.0$ |
| Top-20 Accuracy | $\theta_3 = 0.5$ | 0.8859 | 0.8877 | 0.8815 | 0.8837 |
| | $\theta_3 = 1.0$ | 0.8788 | 0.8850 | 0.8886 | 0.8859 |
| | $\theta_3 = 2.5$ | 0.8797 | 0.8846 | 0.8841 | 0.8819 |
| | $\theta_3 = 5.0$ | 0.8734 | **0.8909** | 0.8824 | 0.8868 |

## 7.6 Weight Setting in Loss Function

As given in Eq. (15), three weight terms are introduced to combine different losses (the POI prediction loss, trajectory forecasting loss, and consistency loss). In this section, we fix $\theta_1 = 1$ and manipulate the remaining two weights ($\theta_2$ and $\theta_3$) to fully investigate the influence of these weights. Both $\theta_2$ and $\theta_3$ are set to one of 0.5, 1.0, 2.5, and 5.0, which results in 16 different combinations. Considering the large amount of the experiments, only the FS-NYC dataset is selected for evaluating. Table 7 reports the POI prediction performance of different combinations on the validation set of FS-NYC dataset. The combination with the best performance for each metric is shown in bold. As it can be seen from the table, the combination of $\theta_2 = 1.0, \theta_3 = 1.0$ achieves top performance in both Top-1 accuracy and Top-5 accuracy, whereas $\theta_2 = 0.5, \theta_3 = 0.5$ and $\theta_2 = 1.0, \theta_3 = 5.0$ is the top performer in Top-10 and Top-20 respectively. Based on these results, in the rest experiments, $\theta_2 = 1.0, \theta_3 = 1.0$ is used as the default setting.

## 7.7 Different Observation Lengths

In the experiments given in the main paper, the observation length $n$ is set to 20. In this part, we focus on investigating the effect of different observation lengths. Figure 4 shows the performance when the observation length is set to $n = 5$, $n = 10$, $n = 20$, and $n = 30$, respectively. As demonstrated from the figure, we can see that the top-5, top-10, and top-20 performance with different observation lengths are quite close. Larger observation length settings ($n = 20$ and $n = 30$) slightly outperform the smaller observation length settings ($n = 5$ and $n = 10$). For the top-1 accuracy, when the observation length is smaller, the performance drops as it is hard to predict when there are only a few visiting records. However, we can also notice that the top-1 performance does not improve when the observation length is too large ($n = 30$). This is because more distant inputs have a relatively smaller contribution (to the next POI) than more up-to-date visiting records.

## 7.8 Model Training Pseudo-code

In Algorithm 1, the pseudo-code of MobTCast training process. Note that we only use user $u_i$ as an example in the pseudo-code for simplification. In the experiments, all users' data in the training set are included for training and the training instances are processed through mini-batches (batch size 512).

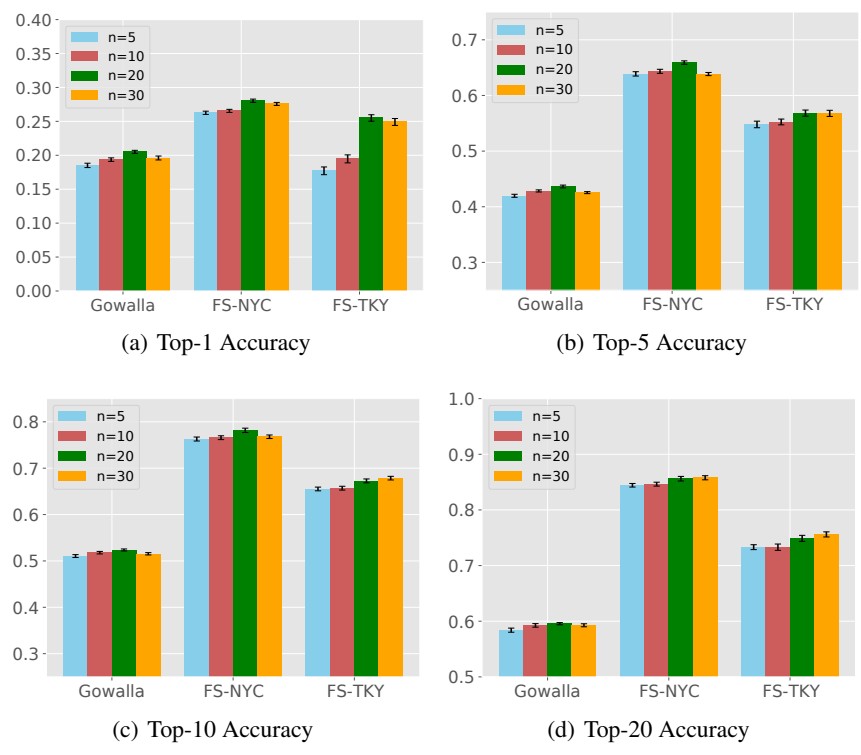

Figure 4: POI prediction performance (on the testing set) with different observation length ($n$) settings.

---

**Algorithm 1** Pseudo-code of training MobTCast (using one user as an example)

---

**Input:** user id: $u_i$; observed sequence of user $i$ (visited POI id): $P_i$ ; observed sequence of user $i$ (timestamp id): $T_i$ ; observed sequence of user $i$ (category id): $C_i$ ; observed sequence of user $i$ (geographical coordinate): $X_i$ ; sequences of neighbours: $\{P_j\}, \{T_j\}, \{C_j\}, j \in \mathcal{N}_i$

**Output:** trained model parameters: $\gamma$

1: $\gamma \leftarrow$ random initialisation
2: **while** not converge **do**
3:      Calculate feature $h^i$ with $P_i, T_i, C_i$ as input by Eqs (1)-(3)      ▷ Semantic-aware mobility feature extraction
4:      **for** $j \in \mathcal{N}_i$ **do**
5:          Calculate feature $h^j$ with $P_j, T_j, C_j$ as input by Eqs (1)-(3)      ▷ Extract neighbour's mobility feature
6:      **end for**
7:      Calculate social context vector $H^i$ with $h^i$ and $h^j$ of each neighbour through Eq. (4)      ▷ Model social influence
8:      Calculate feature $g^i$ with $X_i$ as input by Eqs (5)-(6)      ▷ Auxiliary branch
9:      Embed user id $u_i$ through Eq.(7)      ▷ User embedding
10:     Predict the next location (in the geographical coordinate format) by Eqs. (8)-(9)      ▷ Get prediction output of the auxiliary branch
11:     Predict the next POI by Eq. (10)      ▷ Get prediction output of the main branch
12:     Calculate $\mathcal{L}_i$ with predictions of step 10 and 11 through Eqs (11)-(15)      ▷ Loss calculation
13:     Update $\gamma$ using $\nabla_\gamma \mathcal{L}_i$      ▷ Update model parameters by back propagation
14: **end while**
15: **return** $\gamma$

---