# OpenReview forum: "MobTCast: Leveraging Auxiliary Trajectory Forecasting for Human Mobility Prediction"
_NeurIPS.cc/2021/Conference — NeurIPS 2021 Poster_

### Official Review · Reviewer_5Ds5 · 2021-07-04

**Rating:** 7
**Confidence:** 4

**Summary:**

This paper proposes a Transformer-based context-aware network for human mobility prediction. Specifically, the authors explored the influence of four types of context in mobility prediction: temporal, semantic, social, and geographical contexts. A base mobility feature extractor is designed using the Transformer architecture, which takes both the history POI sequence and the semantic information as input. A self-attention module to model the influence of the social context. Moreover, the authors proposed to use an auxiliary task to model the geographical context and predict the next location. In the experimental results, the proposed method outperforms other state-of-the-art next POI prediction methods and illustrates the value of including different types of context in the next POI prediction.

**Ethical Concerns:**

I do not see major ethical concerns in this paper.

**Limitations And Societal Impact:**

The authors discussed the limitations of the proposed method and provided some insights on the future directions for improving the current method. However, there seems no discussion on the potential societal impact.

**Main Review:**

The paper is generally well written and easy to follow. The detailed comments are listed below.

-- Originality: There seems two major novelty and contribution in this paper. One is to employ an auxiliary trajectory forecasting task to predict the geographical coordinates of the next most likely location. The other is to propose a novel consistency loss function to connect the predicted POI and the predicted location, which further improves the POI prediction performance. I think that these two ideas are interesting and novel. The authors provided an informative literature review on related work and discussed major differences between the proposed method and existing approaches.

-- Quality and Significance: The authors conducted extensive experiments on three widely used datasets and demonstrated promising results, which can support the claims of the authors. The comparison with state-of-the-art baselines and ablation baselines is illustrated with sufficient details.

-- Clarity: The paper is clearly organized and well written.

**Time Spent Reviewing:**

2.0 hours

---

> ### Author Response · Authors · 2021-08-09
> **Response to Reviewer 5Ds5**
>
>
>
> We thank the reviewer for the positive and encouraging review.
>
> Discussion on the potential societal impact: We have briefly discussed the social impact of a human mobility prediction system in the Introduction (Lines 21-26). In addition, as suggested by Reviewer yyia, our work could be valuable in the context of pandemics, to support contact tracing, crowd management and help warn citizens regarding POIs with high incidence levels or recent outbreaks that they are likely to visit (according to our model). We will elaborate further on the societal impact of our work both in the introduction and the discussion parts of the paper. Thank you for your valuable suggestion.

---

> > ### Comment · Reviewer_5Ds5 · 2021-09-01
> > **Thanks for the rebuttal**
> >
> > Thanks for taking the time to write a rebuttal to address my comments.

---

### Official Review · Reviewer_noAt · 2021-07-15

**Rating:** 5
**Confidence:** 5

**Summary:**

In the submitted manuscript, the authors proposed a Transformer-based method to predict the next POI (place-of-interests) of a sequence. The model takes into account not only the temporal sequence of previously visited POIs but also semantic, geographical, and extracted social information. Moreover, an auxiliary trajectory prediction task is introduced to increase the performance of the main task.
To prove the efficacy of the proposed strategy, the model is tested against three publicly available datasets: Gowalla, FS-NYC, and FS-TKY. Thanks to the modeling of the additional context information, MobTCast is able to outperform state-of-the-art results by a 7.22% margin with respect to previously proposed architectures.

**Limitations And Societal Impact:**

A related negative social impact to the POI prediction is the privacy of the users. The work did not discuss how the users' privacy is assured and in what context should this prediction model be used.

**Main Review:**

Novelty:

The main originality of the paper is the usage of the auxiliary task of location prediction in addition to the POI prediction task. Otherwise, either the social/semantic context embedding or the application of transformer encoding are previously used in related works.

Quality:

The paper provides a sufficient description of the proposed technique and the used functions. It also provides an experimental evaluation compared to previous work and ablation experiments to show the importance of the integrated parts in the model. However, the work has some weakness-points:
- Nothing is delivering the performance of the auxiliary task; for example, how accurate is the predicted location or how close is it to the site of the predicted POI.
- It is not clear how the results of the previous works in the comparison are obtained.
- The paper reports a single number (7.22%) of improvement over the three used datasets without specifying the number of top predictions used (top-k). It would be more informative if the gain is computed for each value of k and each dataset separately.
- The model explicitly include four context "types": temporal, semantic, social, and geographical. Following a data-driven approach, social links and neighbors are extracted from visited POIs. However, the same reasoning does not hold for the semantic and geographical context. What if they are not provided? It would be interesting to evaluate the model's robustness when making predictions using inferred semantic classes instead of ground-truth ones.

Clarity:

In general the paper is clear but it contains few writing typos and could benefit of some clarifications:
- Line 270: average -> averaged
- Figure 1 is a bit confusing. It could be improved using a lighter notation and a more symmetrical layout. The legend is also difficult to interpret, and coloured shapes should be put closer to their definitions.

Significance:

From the reported results, the proposed model has noticeable an improvement over the compared previous work, but some clarification is required for how they obtained the previous work results.

**Time Spent Reviewing:**

7

---

> ### Author Response · Authors · 2021-08-09
> **Response to Reviewer noAt**
>
> We appreciate and are grateful for the reviewer’s constructive and valuable feedback. Below, we address their comments and questions.
>
> 1. The performance of the auxiliary task:
> The discussion of the performance of only using the auxiliary task for the next POI prediction is included in section A.4 (Auxiliary Task for POI Prediction) in the supplementary material. Based on the results listed in Table 5, only using the auxiliary branch to predict the next POI does not yield good prediction performance. Due to space limitations, we could not include the analysis in the main manuscript.
> Since the core task of MobTCast is to predict the next POI Id, the performance of the auxiliary task (generating the predicted location) is not reported in the submitted version. Here, we load the trained weights of MobTCast for each dataset and yield the output of the auxiliary branch of MobTCast as the predicted location to be compared with the ground-truth location of the next POI. The average RMSE between the predicted location and the ground-truth location is: 60.202 (Gowalla), 0.096 (FS-NYC), 0.073 (FS-TKY). The error of Gowalla is larger than the other two datasets. This is because Gowalla covers a very large region (POIs across multiple cities), whereas FS-NYC/FS-TKY is only for a single city.
> These results will be added to the supplementary material in the final version of the paper. Thank you for the suggestion.
> 2. How the previous work results are obtained:
> Note that the source code of the methods from previous work are available on GitHub. We slightly modified their implementations (only in supporting functions like data loading and evaluation metrics) in order to use the same experimental setting as ours. During the experiments, we ran all the methods with the same settings (e.g., observation length, train/val/test splitting) as our proposed method to ensure the fairness of the comparison.
> 3. Separate performance gain:
> For the Gowalla dataset, the performance gain (top-1/top-5/top-10/top-20) is: 13.38%/11.38%/8.72%/7.63% (10.28% on average). Similarly, FS-NYC is:1.78%/8.24%/5.24%/1.94% (4.3% on average) and FS-TKY is 10.73%/6.6%/5.99%/5.75%/ (7.08% on average). Note how the most significant improvement is obtained on the Gowalla dataset, which is the most challenging dataset to predict as it has many more POIs and users than the other datasets (see Table 3).
> We will include the above separate performance gain figures in the revised version of the paper.
> 4. Inferred semantic classes:
> As given in A.1 Details of Datasets (Line 559), the semantic category information of the Gowalla is not provided in the original dataset. In our experiments, we inferred the semantic classes using ArcGIS (decoding semantic categories based on the geo-location of each POI). The ground-truth for the semantic classes in the FS-NYC and FS-TKY datasets are provided. Note that MobTCast obtains good performance on the three datasets: the one with inferred semantic classes and the two datasets where the semantic classes are provided.
> To further compare the impact of using inferred semantic classes vs using the provided ones, we took the FS-NYC dataset as an example and used the ArcGIS method to infer the semantic classes in this dataset. The performance (top-1/top-5/top-10/top-20) in this case is: 0.2806(0.0017)/0.6567(0.0037)/0.7769(0.0028)/0.8552(0.0048). The performance of using ground-truth classes (results reported in Table 1) is: 0.2804(0.0024)/0.6591(0.0031)/0.7816(0.0047)/0.8561(0.0041). As we can see, the performance in both cases is very similar, which illustrates the robustness of MobTCast regarding the availability of the semantic classes.
> We did not consider a situation where the geographical context is not provided. To the best of our knowledge, the geo-location of each POI is a fundamental property in current POI datasets (e.g., Gowalla, Foursquare). It would be really hard to infer the geographical context (i.e., GPS coordinates) if no relevant information is provided.
> 5. Clarity:
> Thanks for the very valuable and appropriate suggestions which will undoubtedly strengthen our submission. Hence, we will include them in the final version of our manuscript.
> 6. Societal Impact:
> In this paper, we leverage publicly available datasets (Gowalla and Foursquare) that have been extensively used in previous work [3, 9, 11, 13, 26, 28, 29, 33, 41, 43]. In addition, as stated in Line 282, no personally identifiable information is included in these datasets. We will add a comment in the revised paper, noting that in real-world settings, privacy-preservation could be achieved by existing strategies such as anonymisation, differential privacy [Cynthia Dwork. 2011. Differential privacy. Encyclopedia of Cryptography and Security (2011), 338–340], and also privacy-aware federated learning [Brendan McMahan, et al. 2017. Communication-Efficient Learning of Deep Networks from Decentralized Data. In Artificial Intelligence and Statistics. 1273–1282].

---

### Official Review · Reviewer_z4Tm · 2021-07-16

**Rating:** 6
**Confidence:** 4

**Summary:**

This paper proposed a novel deep learning-based architecture for the next POI prediction. The proposed approach models the temporal and semantic contexts through Transformer, extracts the social context through self-attention module, and model the geographical context via an auxiliary task. Experiments show that it can achieve better results than the SOTA next POI prediction.

**Ethical Concerns:**

No concerns.

**Limitations And Societal Impact:**

Limitations

1. Some important details of model training are missing. For example, given the social context extractor is using the mobility feature extractor, how do we train them together? Is it possible that if we don't train them together from scratch, we could get get better results? Say, in the first several iterations, we train the mobility feature extractor first and then start from the well-trained mobility feature extractor to train the full model. Basically there are connections among different modules in the model, what's the best training recipe?

2. What's the computation cost of the proposed approach, and compared with other approaches?


**Main Review:**

Originality:
This paper has novel design in the POI prediction model architecture. The key different is that the geographical context is captured through an auxiliary task instead of the manually designed kernels.

Quality:
The technique is somewhat sound. The experiments are quite comprehensive, it compares the proposed approach with other 6 approaches on 3 different datasets.

Clarity:
The paper is well-written, most of the points are clearly addressed. Refer to the limitation comments for some points that are not well addressed.

Significance:
This paper seems to address a difficult task in a better way than the previous work. Future works can compare this work as a new POI prediction baseline.

**Time Spent Reviewing:**

2

---

> ### Author Response · Authors · 2021-08-09
> **Response to Reviewer z4Tm**
>
> We thank the reviewer for the constructive feedback and the positive review. Below, we answer the questions raised in the review.
> 1. In MobTCast, the entire model is trained together in an end-to-end fashion. We will add the above sentence in the final version to make this clearer. The current network structure (Figure 1) does not support training different modules separately (e.g., train the mobility feature extractor first and then train the entire model). This is because the outputs from the social feature extractor and the auxiliary location transformer encoder are concatenated before prediction. It is thus not possible to acquire a predicted output (for loss calculation in the training) without training the social feature extractor. In order to train the mobility feature extractor first, a separate pipeline would need to be designed first (e.g., only using the output of the mobility feature extractor to predict). Thus, in MobTCast the entire model is trained together. Exploring the pre-training techniques for the next POI prediction task is a potentially interesting direction for future work.
> 2. Due to the space limitations in the paper, a comparison regarding the computational costs of different models is included in Section A.5 of the supplementary material. Table 6 depicts a comparison of the computational cost of MobTCast with other state-of-the-art approaches and also with variants of our proposed approach. Based on the results shown in the Table, the MobTCast’s inference time is faster than STAN’s but slower than that of other baseline methods such as FlashBack. Comparing the inference time of different MobTCast variants, we observe that the largest contributor to the inference time is the social context, which makes intuitive sense.

---

### Official Review · Reviewer_yyia · 2021-07-22

**Rating:** 6
**Confidence:** 5

**Summary:**

Summary:
The paper focus on building a Point-of-interest (POI) predictor based on temporal, semantic, social and geographical contexts fitted into a deep learning model with aim of overcoming sparsity issue in mobility data. Datasets used are publicly available and results, ablation studies supports the method's efficacy. This is an application paper and has shared code in zip.

Claimed contributions:
1. Auxiliary trajectory forecasting is incorporated in a POI prediction model as a first.
2. A novel consistency loss function to connect the predicted POI (from the main task) and the predicted location (from the auxiliary task).


**Ethical Concerns:**

The plagiarism score as per Docoloc is 12% including references - which is acceptable. Check here:- https://www.docoloc.de/d88c4d5162dd4b21eeb6add446e2b5bbxto_4JX2Nc%7EfbDlilc/en/konto.hhtml?dogetresult=53


**Limitations And Societal Impact:**

There is no negative impact per se, apart from privacy issue of mobility data that may not be fully anonymized following k-anonymity logic. In terms of current pandemic scenario, this work may find some value in application across contact tracing and crowd management - hence has a social impact side.

**Main Review:**


In general, the work is encouraged, however the paper in its current shape needs some overhaul specifically towards scientific contribution.

Strengths:
1. Code, though small sized is available online.
2. Section 4.4
3. Experiments and Supplementary material.

Relevance:
Relevant to NeurIPS application track.

Novelty:
Loss L3 for training MobTCast is sound.

Impact:
On prediction of crowds (useful in pandemic) and also tour planners possibly.

Weaknesses:
1. Writing the paper and linking and describing sections need improvement.
2. Lat, Lng and distances are based on certain geometric models of earth as earth's shape is Geoid. Google Maps uses one, Satellite, Navy, Flights others. What system is adhered here is not clear and have not been thought of it seems. What happens if POI spans across large regions and small regions?
3. Why can't this problem be modeled as a recommender problem? Justify.
4. How are timestamps validated across time-zones? Are they all in GMT or offset taken care of?
5. Analysis is missing from Experimental section.

Correctness:
1. Line 51 - "For example, people are more likely to visit Food POIs at lunch time than at other times. " - should not this be temporal context?
2. Line 58 - "manually designed functions" - usually human designed functions wrt semantics are better than those learnt by automation? Defend.
3. Line 184 - something works here do not mean it will work somewhere. An easy example to relate - deep learning works well for images, fails miserably for signals.
4. Line 216 - different recommender problems have their own inherent characteristic. Like music has genre and shopping has market basket influence.
5. Line 246 - extra softmax layer not clear.
6. Line 293: why Adam optimizer?

Effort: Effort has been made more in writing the paper more than that put in coding and thought process (philosophy). Reverse was expected.

Suggestions:
0. Give an example to illustrate the problem.
1. This is a technical paper venue. Your abstract should contain numbers instead of promising results. Directly jump to the point and results and the philosophy. Elaboration can be done in Introduction.
2. The long sentences should be broken into small sentences for clarity, continuity and smooth reading. May be another round of proof reading will enhance the grammatical and contextual use of words.
3. It is recommended to give a diagram in 'Introduction' section to make sense of the problem statement and application at a glimpse.
4. May be related work gaps can be enlisted in a table and what extra / new has been done be highlighted, focusing on whether new work is at all needed or just a feature nice to have?
5. Overcome repetitions - there exists many text not needed again - focus more on the technical content.
6. Make contribution section crisp. You have introduced novel method - so what? Why others will use yours instead of their own? Highlight this in your preferred way.
7. De-clutter useless information - Line 130 - For example, in computer vision, it has 131 been used for semantic segmentation [25, 5], crowd counting [45], depth completion [27], and visual localisation [34]. - wastage of references and text area.
8. Follow Train-Eval-Split following Andrew Ng's ML Yearning book.
9. How does demographics and world events bias the POI prediction? A scope of future work.


**Time Spent Reviewing:**

4

---

> ### Author Response · Authors · 2021-08-09
> **Response to Reviewer yyia**
>
> We thank the reviewer for the very valuable, detailed and constructive feedback. Below, we include an answer to the provided comments, questions and suggestions. We hope that the reviewer will find the answers below according to expectations. Thank you!
>
> Weaknesses:
> 1. We will implement the necessary changes in the paper to better connect the different sections as per your suggestion.
>
> 2. The geolocation of each POI provided by the datasets are latitude and longitude GPS coordinates. While this is mentioned in A.2, Line 578, we will clarify the geolocation format in the main paper. As stated in Line 293, before using in the auxiliary trajectory forecasting in MobTCast, each coordinate is normalized to a [-1, 1] range (min-max normalization) for each dataset. As for the distance, the only related operation in MobTCast is the loss in the auxiliary task (Eq.(12)). For this loss function, we use the normalized values (e.g., the ground-truth location of the next POI).
> In the case of the Gowalla dataset, the included POIs are across different cities, whereas FS-NYC/FS-TKY are only about a single city. As compared in Table 1, the proposed MobTCast works well on all three datasets (relatively large Gowalla and small FS-TKY/FS-NYC), which indicates that the proposed method is robust enough to work well on both large and small region scales.
> We will add this explanation in A.1 Details of Datasets to make it clearer.
>
> 3. The overall next POI prediction task can indeed be formulated as a recommendation problem. We assume that this comment refers to the auxiliary trajectory forecasting task being modelled as a location prediction problem instead of a recommender problem. Existing recommender systems focus on predicting the next POIs from the perspective of user preferences. However, in this paper, we aim to predict the next POI from the perspective of human mobility by leveraging an auxiliary task to predict the future location. The goal of auxiliary trajectory forecasting is different from top-k recommendation, as it is predicting the next location and treating location data as continuous values.
>
>
> 4. The timestamps and time zones in the different datasets depend on the dataset. The Gowalla dataset provides a column called “check-in time”. However, it does not specify whether it is GMT or the local time. The FS-TKY and FS-NYC datasets include both GMT and time zone offset for each check-in. So, they can be converted to local times. MobTCast supports both GMT and local times, as long as the time system used is consistent between training and inference, i.e. the same method (GMT or local time) is used for both training and inference.
>
> 5. In our paper, the analysis of the experimental results are included under Experiments. Specifically,  Section 5.1 contains the analysis related to model performance. The second paragraph of Section 5.2 contains the analysis regarding the ablation study. Section A.4 and A.5 cover the analysis of the auxiliary task and the computation cost. In addition, the analysis of the datasets and a data-driven analysis of the motivation of incorporating different contexts are included in Section A.1 and A.2.
>
> Correctness:
> 1. We agree that this sentence is ambiguous in its current format. We will update it to “At lunch time, people are more likely to visit Food POIs than other types of POIs.” This example illustrates how the semantic and temporal contexts are often related to each other, which justifies the operation of concatenating the temporal and the semantic embeddings (Eq.(1-3)) in the mobility feature extractor in our MobTCast.
>
> 2. Lines 57-62 point out that one limitation of existing work is that of using manually designed functions to model geographical context. These functions are often based on calculating the distance between two POIs. When the total number of POIs is large, this requires pre-computing all the pairwise distances between every two POIs for model training, which is computationally costly. MobTCast does not require such calculations as the influence of the geographical context will be learned in the training phase. In addition, the proposed auxiliary trajectory forecasting --which includes the geographical context-- has been shown to be effective to predict the next POI. We hope that our work will inspire future work to further explore how to automatically incorporate the geographical context in the models, instead of using handcrafted distance-based features.
>
> 3. Indeed, deep learning-based models are not necessarily the optimal type of model in every task. However, note that the examples given in Line 184 are all time-series modelling tasks. The prediction of the next POI is also a time-series problem. Thus, it seems reasonable for a Transformer to be a suitable architecture in MobTCast, as shown in previous work (see e.g. Wu, N., Green, B., Ben, X., & O'Banion, S. (2020). Deep transformer models for time series forecasting: The influenza prevalence case. arXiv preprint arXiv:2001.08317).
>
> 4. Agreed. For the next POI recommendation, the inherent characteristic is the geographical context (Line216), which inspires us to specifically design an auxiliary branch for MobTCast.
>
> 5. The next POI prediction task is modelled as a classification task (given in Line 252). The softmax layer (after $\text{MLP}_f$ in Figure 1) is introduced as the last layer in the POI prediction (the main branch in MobTCast) so that the output is a vector (whose dimension equals to the total number of POIs) representing the probability of visiting each POI.
>
> 6. The Adam optimizer is a standard and effective optimizer for training deep learning models and it is also used for both the next POI prediction and the trajectory forecasting tasks. It has been widely used in previous work, such as [3, 9, 20, 28, 29, 32, 33, 41].
>
> Suggestions:
>
> Thanks for these valuable suggestions which will certainly strengthen our submission. We will include them in the revised version of the paper. For suggestions 3 and 4, we will add the recommended figure/table if the space limitations allow. In fact, there should be more space in the paper after applying suggestions 5 and 7. For suggestion 8, it would be difficult to address it in the final version as it requires redoing all the experiments. Note that although a different train/val/test splitting is applied, the comparison of different methods is still a fair comparison as the same train/val/test was used for all methods.

---

### Decision · Program_Chairs · 2021-09-27

**Decision:**

Accept (Poster)

**Comment:**

This paper proposes a transformer-based predictor for future places of interest. Its initial scores were 7, 6, 6, and 5; and they did not change during the discussion. The paper presents good empirical results. However, the reviewers were concerned that it is not very technically deep. I read the paper, and see both positives and negatives. Let me start with the positives:

1. Experiments in Table 1 are impressive. A lot of variability in baselines. Three datasets and top-$K$ for various $K$.

2. The ablation study in Table 2 shows that MobTCast performs well even without social context, and losses $L_2$ and $L_3$. In short, the loss $L_1$ is enough to be comparable or better than the state of the art. This indicates that the improvement is due to better sequence modeling using the transformer.

Now the negatives:

1. I am not sure what is the lesson learned from this paper. How would somebody who uses transformers benefit from this result? The paper is indeed not very deep.

2. I am not 100% sure that MobTCast improves over the state of the art because of the transformer. In particular, do any of the baselines use the semantic context in Section 4.1? If not, the observed improvement may be simply because of better features.

In my opinion, this paper should be judged as an application paper, and it clears the bar because of good empirical results. I strongly suggest that the authors take my comments into account, in addition to the detailed comments of the reviewers. Congratulations!